# Effects of Different Drought Degrees on Physiological Characteristics and Endogenous Hormones of Soybean

**DOI:** 10.3390/plants11172282

**Published:** 2022-08-31

**Authors:** Qi Zhou, Yongping Li, Xiaojing Wang, Chao Yan, Chunmei Ma, Jun Liu, Shoukun Dong

**Affiliations:** 1Agricultural College, Northeast Agricultural University, Harbin 150030, China; 2Institute of Crop Science, Chinese Academy of Agricultural Sciences, Beijing 100081, China

**Keywords:** soybean, physiological characteristics, endogenous hormones, drought resistance

## Abstract

Drought affects crop developmentnand growth. To explore the physiological effects of drought stress on soybean, HeiNong44 (HN44) and HeiNong65 (HN65) varieties were used as experimental materials and PEG-6000 was used as the osmotic medium. The antioxidant enzyme activity, osmotic adjustment substance content, antioxidant capacity, and endogenous hormone content of the two soybean varieties were studied under different drought degrees and different treatment durations. Drought stress caused significant physiological changes in soybean. The antioxidant enzyme activities, osmoregulation substance content, and total antioxidant capacity (T-AOC) of HN65 and HN44 showed an increasing trend under mild and moderate drought, however, they first increased and then decreased under severe drought conditions. Following the extension of treatment time, malondialdehyde (MDA) showed an increasing trend. As drought increased, gibberellin (GA) content showed a decreasing trend, while abscisic acid (ABA), salicylic acid (SA), and zeatin nucleoside (ZA) content showed an increasing trend. The auxin (IAA) content of the two varieties showed opposite change trends. In short, drought had a significant impact on the physiology of these two soybean varieties; however, overall, the drought resistance of HN65 was lower than that of HN44. This study provides a research theoretical basis for addressing the drought resistance mechanism and the breeding of drought resistant soybean varieties.

## 1. Introduction

Soybean (*Glycine max* L. Merr.), a dicotyledonous legume, originated in the Yellow River Basin in northern China and is one of the most important economic crops worldwide. The cultivated soybean used today evolved from wild soybeans *(Glycine soja Sieb. & Zucc.)* 6000–9000 years ago [1]. As the fourth most important crop in the world, soybean is a major source of high-quality edible protein for human beings and one of the top-grade feeds for livestock [2]. In 2019, the global yield of soybean exceeded 300 million tons. Presently, Brazil, the United States, and Argentina are the main producers of soybean, and their yield accounts for 34.25%, 29.01%, and 16.58%, respectively, of the total soybean production worldwide [3]. Although the planting scale and yield of soybean are huge, the yield of soybean cannot meet current demand. Extreme climate events pose a considerable threat to the yield and quality of soybean. Particularly, drought severely affects soybean yield, and can reduce soybean yield by 40% in grave cases.

Drought and other abiotic stresses have significant effects on crop physiological activities. For example, drought reduces leaf area, leaf water potential and photosynthetic rate, slows down the speed of material transport, and so on [4]. Moreover, drought induces a sharp increase of ROS content, resulting in oxidative damage, and may lead to plant death in severe circumstances [5,6]. To effectively reduce the production of ROS, plants have evolved complex and precise antioxidant systems. Antioxidant enzymes are a crucial component of the antioxidant mechanism, mainly including catalase (CAT), peroxidase (POD), superoxide dismutase (SOD), and ascorbate peroxidase (APX), which play a vital role in the scavenging of reactive oxygen species [7].

In addition to increasing antioxidant enzyme activity, the accumulation of osmotic adjustment substances such as soluble protein (SPs), soluble sugar (SSs), and proline (Pro) is one of the important strategies used by plants to resist drought stress [8]. Pro and other osmotic regulators can reduce membrane permeability and play an vital role in maintaining intracellular moisture balance in plants under drought stress [9]. Pro is a key participant in plant tolerance to abiotic stress; it can regulate cell osmotic potential and has multiple functions, such as scavenging free radicals under stress, buffering redox potential, stabilizing cell structure, and activating cell pathways [10].

Total antioxidant capacity (T-AOC) is an important indicator of the overall antioxidant level of plants based on various antioxidant enzymes and antioxidants, and can quickly and reliably screen out crop varieties with high drought resistance [11]. In addition to the total antioxidant capacity, the concentrations of malondialdehyde (MDA) and plant hormones are important indicators of the drought resistance of plants. Generally, the higher the drought degree, the higher the MDA content in plants. Therefore, MDA content is used as a marker of oxidative lipid damage to reflect the response of plants to stress [7,12]. As a crucial signaling molecule, plant hormones resist or adapt to drought stress by jointly responding to water deficit under drought stress [13]. Under drought conditions, plant hormones such as auxin (IAA), abscisic acid (ABA), and gibberellin (GA) play a crucial role in crop growth and yield [14]. For example, ABA can significantly reduce stomatal conductance [15], ZA has an important effect on grain formation [14], SA can effectively reduce the oxidative damage to plants [16], IAA is crucial for the formation of crop yield at flowering stage under drought conditions [14], and the decrease of GA concentration slows down the growth rate of crops, which can effectively alleviate the damage caused by drought [17].

Two common methods are used to simulate drought stress in the experiment: not watering and the use of osmotic adjustment substances (such as PEG-6000) to regulate osmotic potential [18,19]. The use of osmotic adjustment substances controls water content in a stable drought state more easily than non-watering [20]. Therefore, in order to make all pots under stable drought conditions, PEG-6000 was used to simulate drought stress in this experiment. Moreover, the selection of varieties with different stress resistances in current research on plant stress resistance can verify results for simultaneous research and explain the differences in stress resistance among varieties [21,22]. Therefore, we selected drought-resistant variety HN44 and sensitive variety HN65 as experimental materials to explore the changes in antioxidant capacity, osmotic adjustment ability, and endogenous hormones in soybean leaves under different drought degrees. Our experiment aims to provide a theoretical basis for research on soybean cultivation in arid areas and the breeding of drought-resistant varieties.

## 2. Results

### 2.1. Effects of Drought Stress on Antioxidant Enzymes of Soybean

As shown in Figure 1, the relative water content (RWC) of soybean leaves changed significantly under different drought degrees. During the seven days of treatment, the RWC of soybean leaves did not change significantly under CK conditions, and the RWC of HN44 leaves was basically maintained at about 75%. The RWC of HN65 leaves was slightly higher than that of HN44, which was maintained at about 80% in general. Under LD conditions, the RWC of soybean leaves remained flat first, then decreased and maintained at a stable state. HN44 decreased significantly on the 5th day and HN65 decreased significantly on the 3rd day. Under MD conditions, the RWC of HN44 leaves remined flat at first, then decreased, decreasing significantly on the 5th day. The RWC of HN65 leaves showed a decreasing trend. By day 7, HN44 was 14.34% lower than that of the CK and HN65 was 26.27% lower than that of the CK. Under SD conditions, the RWC of HN44 leaves remained flat at first, then decreased, decreasing significantly on the 3rd day. The RWC of HN65 leaves showed a continuous downward trend. By day 7, HN44 was 42.55% lower than CK, HN65 was 67.92% lower than CK.

### 2.2. Effects of Drought Stress on Antioxidant Enzymes of Soybean

As shown in Figure 2, different drought degrees had significant influence on the antioxidant enzyme activity of HN44. The results shows that in the CK condition, the activities of CAT, POD, SOD, and APX did not change significantly over the 7 days of treatment. Under LD and MD conditions, the antioxidant enzyme activity of HN44 increased continuously as the treatment time increased. On the 7th day, the SOD activities were 122.67% and 167.86% higher than that of the CK, respectively; POD activities were 335.56% and 435.24% higher than that of the CK, respectively; CAT activities were 255.48% and 390.40% higher than that of the CK, respectively; and APX activities were 77.53% and 179.34% higher than that of the CK, respectively. In SD treatment, following the extension of treatment time, the activities of CAT, POD, SOD, and APX of HN44 increased initially and then decreased, reaching the maximum on the 5th day. On the 7th day of SD treatment, antioxidant enzyme activities were significantly decreased.

Figure 2 shows that PEG-induced drought stress also had significant impact on the antioxidant enzyme activity of HN65. As with HN44, the activities of CAT, POD, SOD, and APX in HN65 did not change significantly within 7 days under CK conditions. As treatment time increased, the antioxidant enzyme activities of HN65 under LD and MD conditions increased continuously. On the 7th day, the SOD activities were 123.14% and 177.73% higher than that of the CK, respectively; POD activities were 344.29% and 421.17% higher than that of the CK, respectively; CAT activities were 310.43% and 426.09% higher than that of the CK, respectively; the activities of APX were 79.05% and 176.49% higher than that of the CK, respectively. As treatment time increased, the activities of SOD, POD, CAT, and APX of HN65 in SD conditions increased initially and then decreased, and reached the peak on the 5th day of treatment. On the 7th day of SD treatment, SOD activity (compared with the 5th day) decreased significantly, falling even lower than CK, POD, CAT, and APX activity (compared to the 5th day), but still remained higher than that of the CK.

Generally, the activities of CAT, POD, SOD, and APX of HN44 were 14.09 μmol/min/g FW (7.02%), 14.97 μmol/min/g FW (4.80%), 68.83 μmol/min/g FW (8.62%), and 20.72 μmol/min/g FW (12.72%) higher than those of HN65 on the fifth day of SD treatment, respectively. On the 7th day of drought stress treatment, when the antioxidant enzyme system was damaged, the antioxidant enzyme activity of HN65 was constantly lower than that of HN44, and reached the significant indigenous level. The CAT, POD, SOD, and APX activities of HN44 were 15.97%, 23.01%, 76.52%, and 15.87% higher than those of HN65, respectively, indicating that HN44 had strong drought resistance.

### 2.3. Effects of Drought Stress on Osmotic Regulators of Soybean

The results showed that PEG-induced drought stress had a significant effect on the content of osmotic adjustment substances in HN44. Figure 3 shows that the contents of Pro, SSs, and SPs did not change significantly in the 7 days of treatment under CK conditions. Under LD and MD conditions, as treatment time was increased, the content of osmotic adjustment substances in HN44 increased continuously. On the 7th day, the content of Pro was 297.31% and 482.69% higher than that of the CK. The contents of SSs were 147.44% and 238.08% higher than that of the CK. The content of SPs was 271.60% and 333.94% higher than that of the CK. Under the SD condition, the content of osmotic adjustment substances of HN44 reached the maximum value on the 5th day of treatment. On the 7th day, the soybean damage due to drought increased significantly, and the content of osmotic adjustment substances decreased sharply.

Figure 3 shows that drought stress induced by PEG has a significant effect on the content of osmotic adjustment substances in HN65. Under the CK condition, the Pro, SSs, and SPs content of HN65 did not change significantly over time. However, the osmoregulation substance content of HN65 under LD and SD conditions increased over time until the end of the 7th day. The content of Pro was 313.51% and 521.57% higher than that of the CK, respectively. The content of SSs was 155.03% and 229.70% higher than that of the CK, respectively. The content of SPs was 299.68% and 400.56% higher than that of the CK, respectively. The contents of Pro, SSs, and SPs of HN65 increased first and then decreased as treatment time under the SD condition increased, and reached the peak on the 5th day. On the 7th day, the content of osmotic adjustment substances of HN65 decreased significantly similar to that of HN44.

In conclusion, the contents of osmolytes in HN44 and HN65 peaked only on the fifth day of SD treatment, and the content of Pro, SSs, and SPs in HN44 was 99.08 μg/g FW (45.28%), 4.12 (1.49%), and 5.03 mg/g FW (3.49%) higher than those of HN65, respectively. On the 7th day of LD, MD, and SD treatment, the Pro content of HN44 was 13.44%, 10.70%, and 154.91% higher than those of HN65, respectively. The SSs content of HN44 was 6.25%, 12.29%, and 24.16% higher than that of HN65, respectively; the SPs content of HN44 was 17.43%, 9.50%, and 12.65% higher than those of HN65. To summarize, under drought-induced damage to osmotic adjustment substances, the osmotic regulation substances of HN65 were lower than those of HN44, and Pro was the most pronounced.

### 2.4. Effects of Drought Stress on MDA and Antioxidant Capacity of Soybean

The MDA contents of HN44 and HN65 under different drought degrees are shown in Figure 4a. The MDA contents of HN44 and HN65 increased as the drought degree and treatment time increased. Under SD condition, the MDA content reached the maximum after 7 days of treatment. The MDA content of HN65 was 411.94 nmol/g FW at SD, which was 2.44% higher than that of HN44. Under drought stress, the MDA content of HN44 did not show significant difference on the 1st day, but showed significant difference on the 3rd day. The MDA content of HN65 under SD treatment was significantly different from other drought degrees on the 1st day, indicating that HN65 was more sensitive to drought stress.

T-AOC of HN44 and HN65 under different drought degrees is shown in Figure 4b. T-AOC increased as the treatment time at LD and MD increased, and reached the maximum value on the 7th day. Under SD condition, the T-AOC of HN44 and HN65 increased initially and then decreased as time increased; they reached the peak on the 5th day of treatment. On the 7th day of SD treatment, the T-AOC values of HN44 and HN65 decreased by 18.50% and 31.51% compared to the 5th day. Only on the 5th day of SD treatment, the T-AOC value of HN44 was slightly lower than that of HN65. In addition, the T-AOC value of HN65 was lower than that of HN44, indicating that the oxidation resistance of HN44 exceeded that of HN65. In other words, HN44 exhibited relatively high drought resistance.

The SASC of HN44 and HN65 under different drought levels is shown in Figure 4c. Under LD conditions, SASC increased as processing time increased, while HN44 and HN65 reached the maximum at 7 days. Under the MD and SD conditions, the SASC of HN44 and HN65 increased initially and then decreased as processing time increased, and reached the maximum on the 3rd day. In the SD condition, the SASC of HN44 and HN65 decreased significantly on the 5th and 7th day. On the 7th day, the SASC of HN44 and HN65 had no significant difference from the control.

The HRSs of HN44 and HN65 under different drought degrees are shown in Figure 4d. Under LD conditions, the HRSs of HN44 and HN65 increased as treatment time increased, reaching the maximum values on the 7th day. The HRS values of HN65 treated with MD and SD increased initially and then decreased as treatment time increased, and reached the maximum on the 5th day. The HRS values of HN44 treated with MD and SD increased initially and then decreased as treatment time increased, and reached the maximum on the 5th and 3rd days, respectively.

Overall, under drought conditions, the MDA content of HN44 was lower than that of HN65, but the antioxidant capacity of HN44 exceeded that of HN65. The T-AOC, SASC, and HRS values under different drought degree treatments showed significant indigenous differences on the 1st day of treatment; the difference in HN65 exceeded that of HN44, indicating that HN65 was more sensitive to drought stress than HN44.

### 2.5. Effect of Drought Stress on Endogenous Hormone Content of Soybean

The above tests on the antioxidant capacity and osmotic adjustment ability of soybean showed that there was no significant difference in soybeans under different drought degrees on the initial days, and SD treatment for 7 days would cause severe oxidative damage to soybean. Therefore, we chose to determine the content of endogenous hormones in soybean leaves on the 5th day. The results are shown in Figure 5. The IAA, ABA, and SA content in HN44 showed an increasing trend, and reached the maximum value at SD, which were 71.10%, 1340.16%, and 68.50% higher than that of the CK, respectively. The ZA content increased initially and then stabilized as the drought degree increased, reaching the maximum value at MD and SD. The GA content showed a decreasing trend as the drought degree increased, and reached the lowest value under the SD condition, which was 45.73% lower than that of the control.

The ABA, SA, and ZA contents of HN65 showed an increasing trend as the drought degree increased. They reached the maximum values under SD conditions, which were 795.78%, 51.85%, and 105.91% higher than those of the CK, respectively. The IAA and GA content showed a decreasing trend as the drought degree increased, and decreased to the lowest in the SD condition, which were 51.77% and 38.55% lower than those of the CK.

Overall, under drought stress, the change trends of HN44 and HN65 in different hormones were not the same. Under different drought degrees, there was no significant difference between the GA content in HN44 and HN65. There was no significant difference between the ABA, SA, and ZA content of the two soybean varieties under the control condition, and the difference gradually became obvious as the drought degree increased. The IAA content in the two soybean leaves showed the opposite trend as the drought degree increased; it increased in HN44 and decreased in HN65. There was no significant difference between the IAA content of the two under LD conditions. In summary, under drought conditions, the content of endogenous hormones HN44 in leaves was generally higher than that of HN65, indicating that HN44 was more drought-resistant.

### 2.6. Principal Component Analysis

To analyze the results and explore the differences between the physiological indexes of the two soybean varieties under drought conditions, we conducted principal component analysis (PCA) on 12 physiological indexes. As shown in Figure 6, PC1 accounted for 68.1% of the total variance of the model, mainly separating the physiological indexes under different drought degrees. PC2 accounted for 13.8% of the total variance of the model, which was mainly related to soybean varieties. PC1 and PC2 explained 81.9% of the data, indicating remarkable results. T-AOC, Pro, SSs, SPs, ZA, ABA, SA, and MDA were positively correlated with the drought degree, and the smaller the angle between the index and the abscissa, the stronger the correlation. GA was negatively correlated with the drought degree, and less correlated with PC2. HRS and SASC had strong positive correlation with PC2. IAA exhibited a strong negative correlation with PC2, and a weak or non-existent correlation with PC1, indicating that there were significant differences among the drought resistance levels of IAA, HRS, and SASC in HN44 and HN65. There were also certain significant differences between the antioxidant capacity and osmotic adjustment content substances of the two varieties.

### 2.7. Correlation Plot

In order to more clearly express the relationship between the various physiological indices and the treatment time, RWC, drought degree and variety, we performed correlation analysis (Figure 7). The results showed that except for SASC, other physiological indexes were significantly correlated with treatment time. All physiological parameters were significantly correlated with RWC. The concentration of PEG-6000 was significantly correlated with POD, CAT, APX, Pro, SP, MDA, T-AOC, and SASC. Varieties were only significantly correlated with T-AOC and HRS.

## 3. Discussion

Under long-term drought stress, the content of ROS will continue to increase due to partial reduction of oxygenmolecules or due to energy transfer to them, which will break the original dynamic balance of ROS in plant cells. The sharp increase of ROS content causes severe oxidative stress injury to plants [7,23]. To alleviate the damage caused by drought, plants produce antioxidants, flavonoids, and secondary metabolites which protect the plant from drought stress damage by detoxifying ROS and promoting protein and amino acid stabilization when plants are expose to stress conditions [24,25]. In their study of broad beans, Dawood et al. [26] found that the activities of CAT, POD, SOD, and APX increased significantly to clear up excessive ROS after 15 days of moderate drought stress. Studies have shown that antioxidant enzymes can effectively alleviate drought stress damage to plants, however, extreme and prolonged stress intensity severely damages the antioxidant enzyme system, resulting in a significant decline in antioxidant enzyme activity [21]. We found that the antioxidant enzyme activity of drought-resistant varieties exceeded that of sensitive varieties under different treatment times and drought degrees, indicating that drought-resistant varieties are highly tolerant to drought stress, which may be partly due to their higher antioxidant enzyme activity, which better reduces the damage caused by oxidative stress [27].

Ozturk et al. [28] believed that osmotic adjustment substances such as Pro, SSs, and SPs could effectively reduce the water potential of plant cells under drought conditions and prevent cell dehydration to ensure normal plant growth. Moreover, as a special osmotic adjustment substance, Pro had special water absorption ability in addition to osmotic adjustment ability, which could effectively alleviate the instantaneous water shortage. Therefore, osmotic adjustment substances can be used as osmotic protection agents for many plants to help those plants adapt to drought conditions. Previous studies on soybean found that the content of osmotic adjustment substances increased as drought stress days and stress levels increased, to maintain normal metabolism [9,29]. However, prolonged stress days and extreme stress degrees destroy the osmotic adjustment system, thereby significantly decreasing the osmotic adjustment material content [30]. These findings are consistent with the changes in osmoregulation substances observed in our experiments at different drought levels.

MDA is an important indicator of the degree of membrane lipid peroxidation [31]. Research suggests that drought can significantly increase MDA content and T-AOC in soybean. Under the same drought degree, the MDA content of drought-resistant varieties was significantly lower than that of sensitive varieties, while T-AOC was significantly higher than that of sensitive varieties [12]; this was consistent with our research results. Ren et al. [32] studied the changes of the superoxide anion radical scavenging ability of peanuts. They found that the SASC of the varieties with drought resistance increased first and then decreased under drought conditions, while the varieties with poor drought resistance showed a decreasing trend. Tan et al. [33] found that drought stress could significantly improve the hydroxyl radical scavenging ability of wheat. In our experiment, SASC and HRS did not show a regular change trend following increased drought degree and time, probably because the drought resistance mechanism, or because the degree and time of the drought treatment of soybean were quite different from those of previous researchers.

Endogenous hormones are important regulators of plant development and growth as well as being trace signaling molecules. They can exert significant physiological effects at very low concentrations. Under drought stress, plants regulate their own growth and development by regulating the content of various endogenous hormones, and minimize the adverse effects of drought stress [13]. ABA is the main plant hormone that triggers short-term responses (such as regulating stomatal opening and closing) and long-term responses (such as changing root structure) under drought stress. It plays a vital role in regulating plant development and growth [34]. Bhusal et al. [35] suggest that drought stress leads to a significant increase of ABA content, and that the higher the degree of drought, the greater the ABA concentration. Under drought conditions, ABA significantly reduced the net photosynthetic rate by reducing the stomatal conductance of plants, which effectively alleviated the damage caused by drought stress. However, certain studies suggested that the stomatal closure and decrease of transpiration rate were not only related to ABA content, but were the result of the combined action of ABA and ZA [36]. Our results showed that drought stress significantly increased ZA and ABA content, and HN44 increased faster than HN65, indicating that HN44 had greater drought resistance than HN65. In this experiment, GA is the only down-regulated plant hormone under drought stress, and its main function is to promote the elongation of plant stem nodes. The decrease of GA content can slow down the growth rate of plants to alleviate the influence of water deficiency on their normal physiological activities, and effectively improve drought resistance [17]. The large accumulation of SA in plants under drought stress protects plants from oxidative stress and increases their tolerance to drought stress [16]. The effect of drought stress on IAA is highly complex and is mainly manifested in the large differences in different organs and different growth periods of plants. Shi et al. [37] found that in the study of Arabidopsis thaliana, the IAA content in Arabidopsis thaliana with strong drought tolerance was higher, while the IAA content in Arabidopsis thaliana with weak drought tolerance was lower. This study found that as drought stress increased, the IAA content of stronger drought resistant soybean HN44 increased, while that of weaker drought resistant soybean HN65 decreased.

Previous studies have shown that drought resistance of HN44 is stronger than that of HN65 [38]. However, the difference between the drought resistance mechanisms of HN44 and HN65 is unclear. First, our experiment shows differences between the physiological characteristics of the two soybeans with different drought stress durations and degrees. Second, under different degrees of drought conditions, the antioxidant capacity and osmotic adjustment ability of HN44 exceeded those of HN65. Particularly, the difference between SOD activity and Pro content of the two soybeans was greater, indicating that the higher HN44 drought resistance was attributed to the high antioxidant capacity and osmotic adjustment material content, which alleviates the damage caused by drought stress to a relatively considerable extent. Finally, the principal component analysis showed that IAA, SASC, and HRS had strong correlations with soybean varieties, indicating that the difference between the drought resistance of HN44 and HN65 was highly correlated with these metabolic processes. To clarify the differences, the molecular and metabolic regulation is analyzed in the next step, and the differences between different varieties is analyzed via multi-omics to provide a theoretical research basis and guidance for soybean drought resistance cultivation and breeding.

## 4. Materials and Methods

### 4.1. Testing Material

The experiment was conducted in Northeast Agricultural University (45°44′ 43.87 N, 126°43′ 50.42 E). In this experiment, two soybean varieties: HeiNong44 (HN44) and HeiNong65 (HN65), were obtained from the soybean research of Heilongjiang Academy of Agricultural Sciences. Previous experiments showed that HN44 is more tolerant to drought than HN65 [38,39].

### 4.2. Experimental Design

The sand culture cultivation technique was used. The sand was washed and dried. Then, 5 kg of sand was placed in a square flowerpot of length (35 cm) × width (25 cm) × height (18 cm) with four circular holes of 0.7 cm in diameter at the bottom. Seeds with the same grain size and fullness were selected in advance for sowing after sand loading. Then, ten soybean seeds were sowed per pot, leaving six seedlings per pot after emergence. Before the true leaf expansion, they were watered with 500 mL of water daily, and with 500 mL Hoagland’s nutrient solution [40] daily after the true leaf expansion, until all bean sprouts grew five leaflets. Four treatments were set in the experiment. A total of 500 mL 5%, 10%, and 20% PEG-6000 nutrient solution was poured every day when soybeans grew to the five-leaf stage to simulate light drought (LD), moderate drought (MD), and severe drought (SD) stress, and the nutrient solution without PEG was used as the control (CK). Each treatment was repeated three times, and each replicate was planted in eight pots for sampling. The treatment time was 7 days, and the samples were collected on the 0th, 1st, 3rd, 5th, and 7th days of the treatment for detection. The samples comprised a mixture of the 2nd and 3rd leaves in 8 potted plants, and only one sample was taken for each plant. The sampling time was 8:00 a.m. to 9:00 a.m. After sampling, the replicate was quickly frozen in liquid nitrogen, then stored in a −80 °C ultra-low temperature refrigerator (Haier, DW-86L828J, Qingdao, Shandong Province, China).

The activities of antioxidant enzymes, malondialdehyde (MDA), antioxidant capacity, and the content of osmotic adjustment substances were measured on the 0th, 1st, 3rd, 5th, and 7th days; endogenous hormones were determined on day 5. Each indicator was repeated three times to ensure accuracy of the data.

### 4.3. Determination of Physiological and Biochemical Indexes

#### 4.3.1. Determination of Relative Moisture Content of Leaves

Determination of the Relative Moisture Content (RWC) of Leaves was made by the BADR Method [41] using the equation RWC [%] = [(FM − DM)/(TM − DM)] × 100, where FM, TM, and DM are the fresh, turgid, and dry masses, respectively. Three leaf discs for each accession plants exposed to drought and corresponding control plants were cut and immediately weighted (FM), then saturated to turgidity by immersing in cold water overnight, briefly dried, weighted (TM), oven-dried at 80 °C for 24 h, and weighted (DM).

#### 4.3.2. Determination of Antioxidant Enzyme Activity

The activity of antioxidant enzymes was determined using the method reported by Wang [42]. 0.1 g frozen samples were ground in an ice bath with 1.0 mL phosphate buffer (pH = 7.8, 0.05 mmol/L) at 4 °C, 10,000× *g* centrifugal for 20 min. the supernatant was enzyme crude extract. The activities of catalase (CAT), peroxidase (POD), superoxide dismutase (SOD), and ascorbate peroxidase (APX) were determined using the potassium permanganate method, guaiacol method, N-blue tetrazolium method, and ultraviolet spectrophotometry, respectively. There were four test tubes (5 mL): two were used for the determination and the other two as the control.

#### 4.3.3. Determination of Osmotic Adjustment Substance Content

The content of osmotic adjustment substances was determined using the method reported by Zhang [43]. The content of soluble sugars (SSs) was determined using the anthrone method. The content of soluble proteins (SPs) was determined using the G-250 coomassie brilliant blue method. A 1 g frozen sample was ground with 1.5 mL 80% ethanol (adding a little quartz sand) in the pre-cooling bowl, and the volume was fixed to 5 mL with 80% ethanol solution. The extract was transferred into the test tube at 80 °C for 20 min. Afterwards, the extract was filtered twice on the filter paper with activated carbon. The filtrate was placed in the test tube with 0.2 times the weight of zeolite oscillation for 5 min. The supernatant was centrifuged at 4 °C for 10 min at 5000× *g*, and the content of proline (Pro) was determined by acid ninhydrin colorimetry.

#### 4.3.4. Determination of MDA Content and Antioxidant Capacity

The content of malondialdehyde (MDA) was determined using the method reported by Zhang [43]. A 0.1 g frozen sample was placed in a precooled mortar, and 0.8 mL methanol was added to an ice bath and ground. The homogenate was placed in a dark environment at 4 °C for 2 h to make it fully react. Finally, it was centrifuged at 4 °C and 12,000× *g* for 20 min. The supernatant was obtained, and 0.025 mg 2,2-diphenyl-1-pyridine (DPPH) was weighed to configure the reaction solution. The T-AOC was determined according to Bobková’s method [44]. Afterwards, 0.1 g frozen samples were placed in a pre-cooling bowl, 1 mL 80% ethanol solution was added for grinding, homogenized into a 2 mL centrifuge tube at 50 °C 220 W under the condition of ultrasonic extraction for 10 min, and finally, in 12,000× *g*, 4 °C under the centrifugal condition for 10 min in supernatant using the pyrogallol method [45] to determine the superoxide anion radical scavenging ability of the sample (SASC). The extraction process of hydroxyl radical scavenging capacity (HRS) was consistent with the determination of superoxide anion radical scavenging capacity, and the determination method was based on visible spectrophotometry [46].

#### 4.3.5. Determination of Endogenous Hormone Content

A 1 g frozen sample was ground in an ice bath, 20 mL of 80% pre-cooling methanol was added, sealed, and placed in a 4 °C refrigerator overnight. The extract was filtered and washed twice with 10 mL methanol. After filtration, it was combined with the extract and evaporated at 40 °C until there was no methanol residue. The remaining water phase was transferred to a triangular flask, extracted and decolorized twice with 30 mL pe-troleum ether. The ether phase was discarded. Then 0.01 g PVPP (crosslinked polyvi-nylpyrrolidone) was added to the ultrasonic for 30 min and filtered. Afterwards, it was extracted with 30 mL ethyl acetate 3 times, combined with the lipid phase, and evaporated at 40 °C. The residue was dissolved with methanol (chromatographically pure) and diluted to 2 mL. The test solution was filtered through a 0.45 μm microporous membrane and stored in a refrigerator at 4 °C. Then, according to the method reported by Wang, the contents of abscisic acid (ABA), zeatin nucleoside (ZA), auxin (IAA), gibberellin (GA), and salicylic acid (SA), were determined by HPLC [42].

#### 4.3.6. Data Analysis

All data were analyzed using IBM SPSS (version 21.0: IBM Corporation, Armonk, NY, USA) for one-way ANOVA, Microsoft Office Excel 2021 (USA) for line chart, and OriginPro2021 (Origin Lab Corp, Northampton, MA, USA) for principal component analysis and correlation plot.

## 5. Conclusions

This experiment was conducted to study the effects of different treatment durations and degrees on the physiology of two spring soybean varieties with different drought resistance levels. Under drought conditions, soybean enhances its survival ability and yield by enhancing the activity of antioxidant enzymes, the content of osmotic regulators and the content of endogenous hormones such as IAA, ABA, and SA. In order to increase soybean yield through external conditions, a number of measures must be taken. These include the selection of soybean varieties that are suitable for local climate and soil conditions, a robust monitoring of the weather, the use of timely measures such as artificial rainfall during dry years, and the use of exogenous hormones to improve the drought resistance of plants.

## Figures and Tables

**Figure 1 plants-11-02282-f001:**
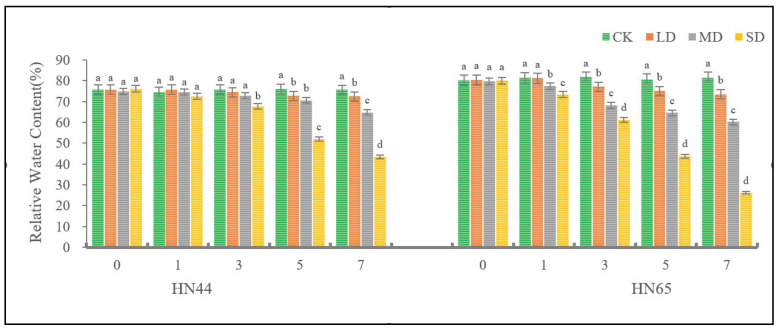
Changes in activities of Relative Water Content of HN44 and HN65 over time under different drought degrees. CK: Control; LD: Light Drought; MD: Moderate Drought; SD: Severe Drought. Different letters in treatments indicate significant differences according to Duncan’s single factor variance test at the 5% level, and data are presented as mean ± SE (standard error) (*n* = 2).

**Figure 2 plants-11-02282-f002:**
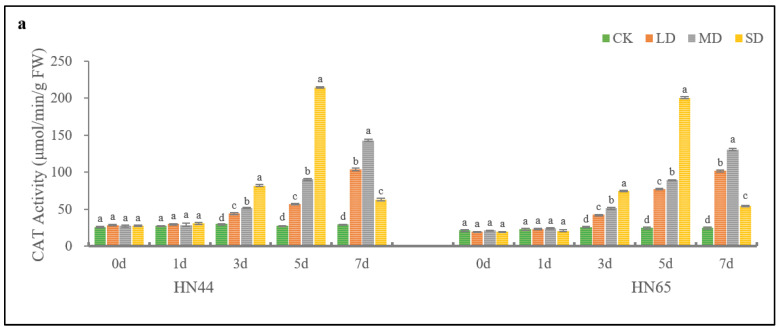
Changes in activities of CAT (**a**), POD (**b**), SOD (**c**), and APX (**d**) of HN44 and HN65 over time under different drought degrees. Different letters in treatments indicate significant differences according to Duncan’s single factor variance test at the 5% level, and data are presented as mean ± SE (standard error) (*n* = 2).

**Figure 3 plants-11-02282-f003:**
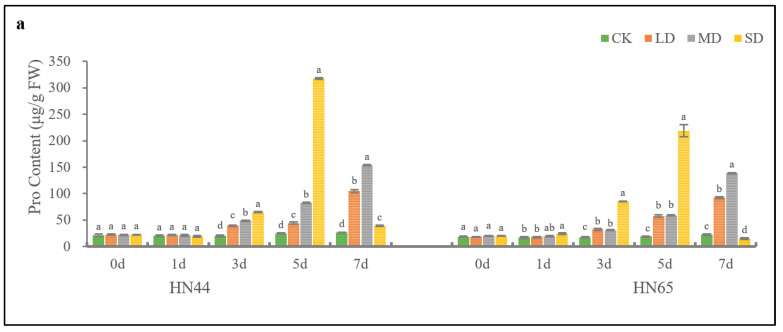
Changes in Pro (**a**), SSs (**b**) and SPs (**c**) contents with time in HN44 and HN65 under different drought levels. Different letters in treatments indicate significant differences according to Duncan’s single factor variance test at the 5% level, and data are presented as mean ± SE (standard error) (*n* = 2).

**Figure 4 plants-11-02282-f004:**
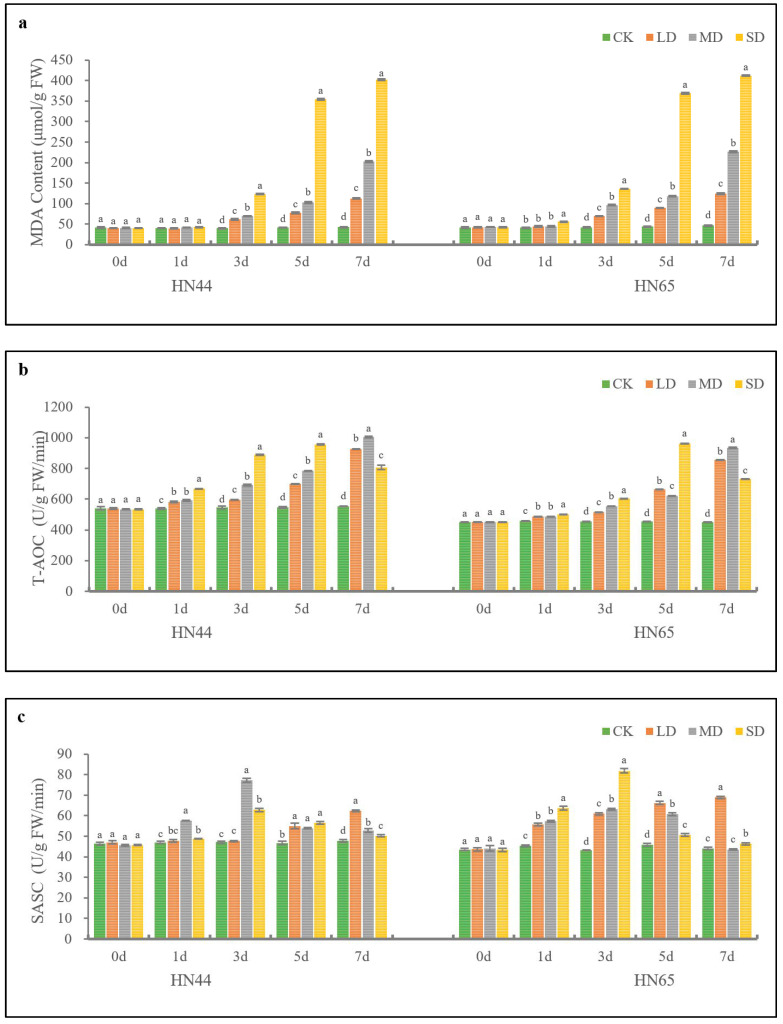
Changes in MDA (**a**), T-AOC (**b**), SASC (**c**), and HRS (**d**) over time in HN44 and HN65 under different drought degrees. Different letters in treatments indicate significant differences according to Duncan’s single factor variance test at the 5% level, and data are presented as mean ± SE (standard error) (*n* = 2).

**Figure 5 plants-11-02282-f005:**
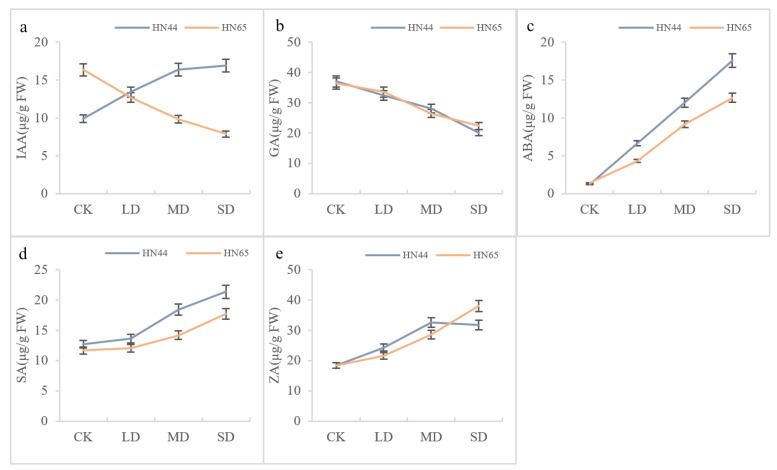
The contents of IAA (**a**), GA (**b**), ABA (**c**), SA (**d**), and ZA (**e**) in HN44 and HN65 under different drought degrees on the 5th day of treatment.

**Figure 6 plants-11-02282-f006:**
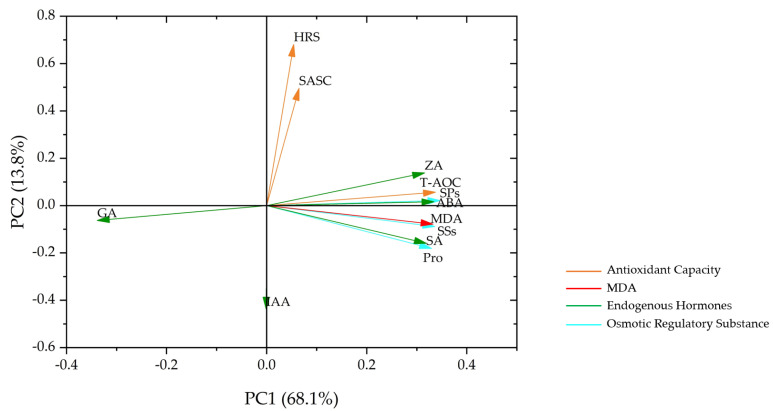
Principal component analysis of physiological indexes on the 5th day of drought showed that orange, red, green, and blue arrows were used to represent antioxidant capacity, MDA, endogenous hormones, and osmotic adjustment substances.

**Figure 7 plants-11-02282-f007:**
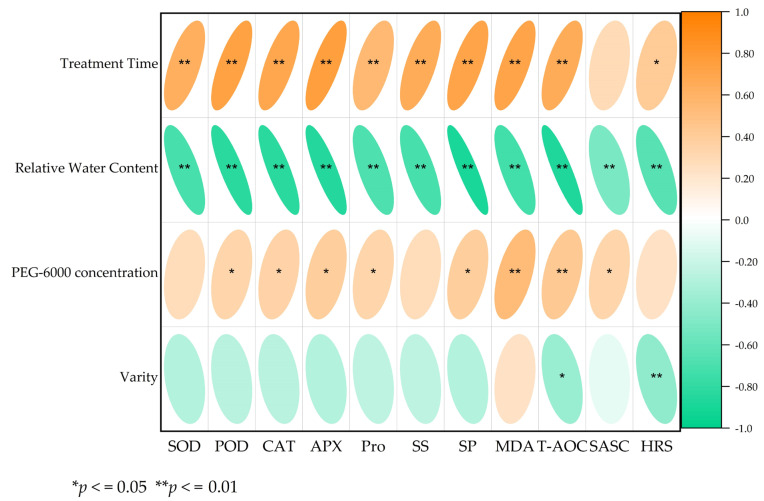
Correlation analysis between different physiological indexes and treatment time, RWC, PEG-6000 concentration and varieties.

## Data Availability

Not applicable.

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
