# Peer review of "Effects of Different Drought Degrees on Physiological Characteristics and Endogenous Hormones of Soybean"

_plants, 2022, doi:10.3390/plants11172282_

Round 1

Reviewer 1 Report

General comments

I have read the manuscript (Plants -1880931). Entitle: Effects of different drought degrees on physiological characteristics and endogenous hormones of soybean written by Qi Zhou et. al., for publication of plant MDPI. In this study, the author investigated the physiological effects of drought stress on soybean used as the osmotic medium and determine the antioxidant enzyme activity, osmotic adjustment substance content, antioxidant capacity, and endogenous hormone of two soybean varieties under different drought stress. In this study author mainly found significant physiological changes and increased the antioxidant enzyme activities, and osmoregulation substance content. Moreover, the author also found that while extending the period of drought stress malondialdehyde (MDA), abscisic acid (ABA), salicylic acid (SA), and zeatin nucleoside (ZA).

The overall research is well conducted, and research is obvious application potential for the readers because this study provides a research theoretical basis for addressing the drought resistance mechanism and the breeding of drought-resistant soybean varieties. In this sense, the manuscript is much valuable. However, I found some points, especially the flow of the text and lack of potential references, and lack of connection of story in different paragraphs, especially in the introduction and discussion sections. The author should provide enough examples and their interpretation of different traits of physiological and biochemical responses by the latest and appropriate references, some of which I mentioned below. Overall after I evaluate and request the author for this manuscript as a “MAJOR REVISION”.

Major suggestions

1)  Introduction: The introduction is well starting with the economic importance of soybean and its overall production scenario on a global scale which is highly appreciated. However, the important message in the introduction especially the effect of drought primarily in the introduction section somewhere in the second paragraph. The article https://doi.org/10.1016/j.foreco.2020.118099” better presented the drought effect on the plant, please follow this article as a reference and mentioned that “drought reduced the morphological and physiological traits, reduce the photosynthesis, leaf water potential and sap movement and reduction of stomatal closure due to mainly closure of the stomatal in the plants”.

   2) Hypothesis and objectives in the introduction: The author should make more clearly present the research hypothesis first and then only its objectives parts secondly. The author should be well connected to these two parts while mentioning the research objective. Please rephrase it slightly from Ln. 68 to 78 for further connection to the hypothesis and its objectives. The hypothesis should be very clear in the introduction sections because, without appropriate literature, questions, or hypotheses in the introduction section the entire text will be unclear. 

3) Discussion: Author should Improve the first paragraph of the discussion more logically with clearer potential references because the main theme of antioxidant and secondary metabolites under drought stress conditions and release the of ROS (why ROS is emerging in stress conditions?). Refer these articles for better clarify (1) /doi.org/10.1038/s41598-019-55889 (2) doi.org/10.1016/j.scitotenv.2021.146466 and mention somewhere in that paragraph “abiotic stress especially environmental stress (I.e. drought) plant produces the ROS when these plant exposed to the stress condition and plant produce antioxidant, flavonoids, and secondary metabolites play to the role for protecting the plant for detoxifying ROS and protect the plant to protect the abnormal condition (i.e. stress) and protein and amino acid stabilization”.

   Some other comments

4) Line no. 319-323 : Generally, when plants are exposed to the drought, the level of ABA is increased, moreover, the higher the drought intensity higher the ABA concentration ABA shows the direct effect by reducing the gs in the plant and stomatal partially or fully close and that result cause the reduction of Pn. The text related to the ABA under different drought intensities is well described in this article https://DOI:10.1016/j.scienta.2018.11.021. This article is good for reference.

5) Line no. 371-372: How author make sure that the concentration of “500 mL 5 %, 10 %, and 20 % 371 PEG-6000 nutrient solution was poured every day, how author prepare this nutrient solution?

6) Line no. 391-392: How author determine the activities of catalase (CAT), peroxidase (POD), superoxide dismutase 391 (SOD), and ascorbate peroxidase (APX), I do not see any methodological description if author follow the others method in here the author should cite their literature.

7) Line no. 422: The author should describe a little more about how to determine the ABA, IAA, GA and SA, it is too short and not enough. Is the author use the HPLC full form before? The author can concise another part inside the manuscript and this section should be more descript.

8) Line no. 430 (Conclusion): The conclusion should not be repetitive in the abstract or a summary of the results section. I would love to read striking points and take-home messages that will linger in the readers’ minds. What is the novelty, how does the study elucidate some questions in this field, and the contributions the paper may offer to the scientific community?

9) Line no. 444 (Reference): please double-check the citations, their style, spell check, and other grammatical errors. moreover, I request to the authors for revision throughout the manuscript according to the journal rules.

 Good Luck!

Author Response

请参阅附件

Reviewer 2 Report

The work presented for review concerns an important issue related to the factors determining the response of soybean varieties to drought. Despite this, in my opinion, the work is not very innovative, basic parameters such as the activity of antioxidant enzymes or phytohormones content have been determined, which has already been tested many times in similar experiments. This is my general comment about this work. I also have some more specific comments:

1) The Introduction contains too general information. For example the issue of the role of individual phytohormones in the plant response to stress can be elaborated.

2) The description of the results is by far the worst written part of the work. Providing the values ​​of individual determinations is a repetition of what can be read from the graphs, it is better to avoid that and focus on describing trends as well as comparing the obtained results between varieties. In addition, there are some errors, eg. lines 226, 234. The sentence on lines 225-227 is incomprehensible. In description of the determination of ZA content, the decrease described in line 227-229 is so slight and statistically insignificant that it can be assumed that the concentration of ZA remains at a similar level at MD and SD.

3) The figures captions are incomplete. There is no explanation of abbreviations, footnotes what is in part a, b, c ... of figures and explanations of what the error bars and letters above the bars indicate.

4) It seems that using one-way ANOVA is not enough to state that there were differences between varieties, two-way analysis would be better.

5) In the experiment drought resistant and sensitive varietes were used, although there is no determination of a physiological parameter that would confirm this, at least relative water content in leaves under the conditions of this experiment. Admittedly, the Authors refer to the work of Wang et al. 2012, but it is not available and it is hard to define on what basis these varieties were classified as drought resistant or sensitive. 

6) In the Discussion (lines 296-307), the Authors compare the obtained results with quite distant plants, such as rice or peony, while there are results obtained for soybean, eg. Dong et al. 2019 or Rao and Chaitanya 2020.

7) Literature should be unified, journal abbreviations should be used.

Author Response

请参阅附件

Round 2

Reviewer 1 Report

Dear Author

I have read the revised manuscript (Plants-1880931). Titled: Effects of different drought degrees on physiological characteristics and endogenous hormones of soybean for publication of Plants MDPI. This is the second submission made by the author. The author addressed all the questions and suggestions that I raised the issue in the review of the original manuscript. I satisfy the author’s revisions throughout the paper. Author well addresses the abstract issues. Especially author improved the introduction and discussion section very well inflow. Now, this manuscript improved the flow of writing, which was comparatively shallow in the original version but in this revised copy author addressed all the quarries and suggestions very well. Before accepting this manuscript, I found small mistakes please correct those accordingly.

1) Author should check the English grammar, or spell check and sentence structure. I strongly suggest that the author should check all the references and should make them consistence writing pattern (e.g., Surname should first always and given name in last)

2) Please correct reference no. 4, the Surname should be in first, and given name should be last.

3) Please correct reference no. 35, the Surname should be first and the given name should be last.

, I request this manuscript is currently in “Minor Revision” I hope the author will improve all the errors in this stage. Thank you.

Reviewer 2 Report

All suggestions from the previous review have been taken into account. After minor spell checking work can be published.

Author Response

Dear Editors and Reviewers:

Thank you for your recognition of our work. We have re-examined the article and made some revisions. The new revision part uses "the tracking modification" function. We hope that our new work can be accepted by you.

Once again, thank you very much for your comments and suggestions.

Qi Zhou

[email protected]